# Growth, Health, Quality, and Production of Onions (*Allium cepa* L.) Inoculated with Systemic Biological Products

**DOI:** 10.3390/microorganisms13040797

**Published:** 2025-03-31

**Authors:** Glenda Margarita Gutiérrez-Benicio, César Leobardo Aguirre-Mancilla, Jesús Manuel Arreola-Tostado, Gerardo Armando Aguado-Santacruz

**Affiliations:** 1Tecnológico Nacional de México/IT de Roque, Km 8 Carretera Celaya–Juventino Rosas, Celaya 38110, Guanajuato, Mexico; 2BIOqualitum, Oriente 7 # 158, Ciudad Industrial, Celaya 38010, Guanajuato, Mexico; 3Campo Experimental Bajío, Instituto Nacional de Investigaciones Forestales, Agrícolas y Pecuarias, Km 6.5 Carretera Celaya-San Miguel de Allende, Celaya 38110, Guanajuato, Mexico

**Keywords:** onion, systemic biofertilizer, systemic biofungicide, yield, quality, endophyte, *Pseudomonas fluorescens*, *Azospirillum* sp.

## Abstract

The efficiency and consistency of biologicals in the field remain a drawback of current conventional products. The development of systemic biological products has opened a new avenue for microbiological and agricultural research. In this study, we evaluated over a two-year period (2022 and 2023) the functioning of two systemic products, a biofertilizer and biofungicide, on the performance of the onion. The first indicator of proper functioning of these products in onions was reflected in increased chlorophyll accumulation. At the end of both years, the inoculated plants were taller, heavier, and developed more leaves than their control counterparts (*p* < 0.5). Inoculated onion bulbs collected at harvest time were significantly heavier (45.1% in 2022 and 56.2% in 2023) than their non-inoculated counterparts (*p* < 0.5). Onion quality, expressed in terms of total soluble solids and pyruvic acid content, was also significantly improved in plants inoculated with the biological products; the two-year average values for these quality variables were 10.2 vs. 14.4°Brix and 2.3 vs. 4.0 µmol∙g^−1^ for control and biologically treated plants, respectively. The two-year average fungal incidence was 1.9 times greater in the control group than in the inoculated plants, while the average onion yield for this period was 44.7% higher (*p* < 0.5) in the biologically treated plot (54.7 t∙ha^−1^) than in the control one (37.8 t∙ha^−1^). Based on comparisons with previous studies employing conventional biologicals, our results demonstrate the superior effectiveness of the systemic biologicas in improving onion performance.

## 1. Introduction

The onion (*Allium cepa* L.) is an economically important vegetable grown worldwide, including Mexico. It belongs to the *Amaryllidaceae* family and is native to Asia [1]. This vegetable is cultivated to obtain fresh and processed products, including medicinal ones. It is a hardy, bulbous winter plant, which is grown annually for bulb production and biennially for seed production. The onion contains carbohydrates, vitamins, minerals, antioxidants, and essential oils [2,3]. Although white, red, and yellow onions are available, the latter make up 75% of all cultivated onions in the world. Onion cultivation is practiced worldwide with an estimated area of 4.5 million hectares and a production of 92.1 million tons; the world average yield is 19.3 tons per hectare [4]. India, China, the USA, Egypt, and Turkey are the top five onion-producing countries in the world [5]. Although India is the largest onion producer, its productivity lags behind as the Republic of Korea has the highest onion productivity in the world (79.6 t∙ha^−1^) followed by the United States (71.1 t∙ha^−1^) [6]. In Uzbekistan, a number of farmers are currently obtaining greater onion yields through the creation of new high-yielding onion varieties suitable for cultivation at different times and the use of improved technologies for growing from seeds, seedlings, and sprouts, in particular, water-saving technologies such as the drip and sprinkler irrigation systems, achieving record yields of up to 150 t∙ha^−1^ [4]. Mexico’s onion production in 2023 was 1,800,000 tons. This country is the world’s leading producer of fresh onions, with this agricultural crop being the fifth most important vegetable in the country. On the other hand, China is the world’s leading producer of dehydrated onions [7]. In Mexico, 80% of the onion crop is grown under irrigation, while the remaining 20% is rainfed. The average yield of onion in this country is around 31 t∙ha^−1^ [8]. Key challenges in onion cultivation in Mexico and worldwide include optimizing fertilization (formula, quantity, timing, and profitability) and implementing effective disease and pest control strategies [9]. To improve the profitability of onion production, farmers have been using high quantities of agrochemicals, especially since the Green Revolution establishment in the late 1960s, when farmers began incorporating new technologies such as high-yielding varieties of cereals and the widespread adoption of chemical fertilizers, pesticides, and controlled irrigation [10].

The sharp rise in the prices of agricultural inputs and the growing interest of society in consuming healthier foods produced with reduced environmental impact have forced the search for new farming alternatives [11]. Consequently, there is growing interest in producing our agricultural foods with minimal amounts of synthetic chemicals considering alternative nutrient sources such as manure, composts, worm fertilizers or biofertilizers, and botanical products or biopesticides to control diseases and pests [11,12]. In this context, the use of biological products is expected to contribute to a more profitable, healthier, and ecological agriculture. Despite the widely reported potential of biological products to increase crop quality and production, reports on the efficacy of biological products are not always conclusive due to the numerous plant and soil factors that affect the functioning of microorganisms in agricultural ecosystems, and issues related to the quality of biological products manufactured [12,13]. Systemic biological products offer a new, more efficient, and powerful alternative to conventional biologicals, because in these products, the microorganisms can penetrate plants via the stomata and reside within the internal tissues of plants, where they are less affected by biotic or abiotic environmental factors. At the same time, the beneficial activities of the microorganisms are more direct, consistent, and effective [11,14,15]. Consequently, systemic biological products have a more significant effect on the production and quality of crops [11,14,15].

This is particularly evident in crops like onions, in which plants have poorly developed and shallow root systems and, consequently, are very sensitive to nutrient scarcity. Moreover, the root system of the plants is ineffective at absorbing immobile nutrients such as P, K, and some micronutrients [16]. Accordingly, onion crops have been shown to respond positively to conventional biofertilizers and organic fertilizers [12,17,18]. Assuming that conventional biological products are manufactured in compliance with basic standards of quality, the consistency and reliability of these products remain a challenge to be solved [13,19,20]. In 2010, our research group developed a novel technology called “*Micro In*”, which enables the introduction of beneficial microorganisms into plants via the stomata, making the employment of microorganisms in agriculture a more efficient and reliable tool for increasing the productivity and quality of crops. Products derived from this technology are called ‘systemic biological products’ [14]. We introduced the term ‘systemic biological products’ in analogy with systemic agrochemicals. For example, in a systemic chemical insecticide, the active compounds applied to plants are absorbed and translocated into other parts of the plants through the vascular system, making these parts toxic to certain insects [21]. A systemic chemical herbicide kills the entire plant by spreading gradually throughout its vascular system, from either foliar application down through the plant, or soil application up towards the leaves [22]. Therefore, as we stated, a systemic biological product allows the introduction of beneficial microorganisms inside plants via plant stomata from where the microorganisms are transported to the vascular bundles and then to the entire plant [11]. Compared with conventional biological products, several advantages of systemic biologicals have been delineated [14]. The interior of plant tissues provides a more favorable environment for the survival and proliferation of plant growth-promoting microorganisms. Additionally, the beneficial activities of these microorganisms are more efficient and consistent because the active compounds that promote growth and health of the plants do not suffer losses (as happens, for example, in the case of nitrogen fixation and hormone production in the rhizosphere) and act more directly on plant metabolism (e.g., hormones, volatile compounds, ACC deaminase, antibiotics, and HCN). Additional advantages of systemic and endophytic microorganisms have been mentioned by other authors [11,23,24]. Therefore, the objective of this research was to evaluate the effects of using two systemic products, a biofertilizer and a biofungicide/biofertilizer, on the health, quality, and yield of onion over two growing cycles.

## 2. Materials and Methods

### 2.1. Study Site

This study was conducted over a two-year period (2022 and 2023) in San Juan de la Vega, Municipality of Celaya, Guanajuato, Mexico (20°39′25″ N; 100°46′22″ W), at an altitude of 1750 m.a.s.l. The soils present at this site are classified as Vertisols [25], which are dark soils with a high content of organic matter and naturally fertile. The white onion cultivar used in this investigation was ‘Carta Blanca’ (a hybrid, short-day variety of onion developed by the INIFAP, National Institute for Forestry, Agriculture and Livestock Research), with a seeding rate of 300,000 plants per hectare spaced 30 cm between rows and 10 cm between plants [26]. The onion seedlings were transplanted from the greenhouse trays to the field in October, while the onion plants were harvested in January in both study years.

In 2022, the annual rainfall was 480.2 mm (51.5 mm during the onion growing cycle), while the average temperature ranged from 15.5 °C in December to 23.4 °C in May. In 2023, the annual rainfall was 453.3 mm (113.5 mm during the onion cycle), and the average temperature ranged from 14.4 °C in December to 24.9 °C in June (Figure 1).

### 2.2. Experimental Treatments

In this research, we evaluated the functioning of two commercial systemic biological products (BactoCROP^®^, a biofertilizer; and Trichonator^®^, a biofungicide/biofertilizer) distributed by the company BIOqualitum (www.bioqualitum.com, accessed on 26 March 2025; patent 328184) on the growth, quality, and yield of the onion (cultivar ‘Carta Blanca’). BactoCROP^®^ contains *Pseudomonas fluorescens*, *Azospirillum brasilense*, and *Bacillus subtilis* as active ingredients (all of them at a minimum concentration of 1 × 10^8^ UFC), whereas Trichonator^®^ contains four different *Trichoderma* species (*Trichoderma viridae*, *T. harzianum*, *T. koningii,* and *T. hamatum*), each formulated at a minimum concentration of 1 × 10^8^ UFC.

These systemic biological products also include minimal concentrations of some other nutrients contained in the organic vehicle employed for their formulation (Table 1). Both products were applied for two consecutive years (2002 and 2023) during the growing cycle of onion. Every year, three applications of BactoCROP^®^ and Trichonator^®^ were used in the one-hectare experimental plot. Both biologicals were additionally applied on the experimental plot in addition to the local conventional crop management; with the exception of the chemical fungicide (methyl thiophanate) normally used by the farmers, which was excluded from the experimental plot. A neighboring (control) plot of the same dimension was managed conventionally, following the complete application program of chemical products normally employed by the local farmers.

The first application of the two biologicals was performed at sowing, when seeds were placed 1.5 cm deep in 338-cavity greenhouse trays; 3.0 kg of BactoCROP^®^ and 1 kg of Trichonator^®^ were dissolved in 600 L of water and then sprinkled on 900 greenhouse trays using a backpack sprayer. The second application was performed at the transplanting time (i.e., approx. 45–60 days after seed sowing) using the same proportion of biologicals for the 900 trays. Finally, the third application was carried out 1.5 to 2 months after transplanting the seedlings to the field; this time, 2.0 kg of BactoCROP^®^ and 750 g of Trichonator^®^ were injected into the drip irrigation system, while 1 kg of the biofertilizer and 250 g of the biofungicide were dissolved in 600 L of water and then sprinkled on the one-hectare experimental plot of the onion using a tractor sprayer.

Inoculation of the systemic biologicals was applied alongside the chemical fertilization regime normally used by the local onion farmers, which consists of applying a 180-80-40 dose (Haifa, Israel) in two fractions. The first fertilization was carried out four to ten days after transplanting, applying half of the N and all of the P_2_O_5_ and the K_2_O (90-80-40). The second fraction of N (90-00-00) was applied 50 days after the first fertilization.

### 2.3. Variables Analyzed

The following variables were determined on both plots (bioinoculated and control) at 60, 90, and 120 days after transplanting (DAT) onion seedling into the field: plant height (cm), fresh weight (g), number of leaves per plant, bulb fresh weight (g), and chlorophyll content (µg∙g FW^−1^). At the end of the onion’s growing cycle, the total yield per hectare (t ha^−1^) and fungal incidence rate (%) were also determined for both the biofertilized and the conventionally managed plots. The incidence rate was calculated with the following formula: Incidence (%) = (number of diseased plants × 100)/total plants observed. The chlorophyll content of the onion leaves was determined according to Aguado-Santacruz et al. [27]. Finally, to evaluate the effects of the systemic biological products on onion quality, the total soluble solids (°Brix) and the pyruvic acid (μmol/g) contents were determined at the end of the productive cycle of the onion. The content of total soluble solids in the bulbs was quantified by using a digital refractometer (model Abbe Leika Mark II, Leica, Solms, Germany) measuring each sample in triplicate and expressing the readings in °Brix. The pyruvic acid content was quantified using the method proposed in [28]. All of the afore mentioned growth and quality variables were determined in 25 samples (seedlings, plant leaves, or bulbs) collected from each treatment. To determine onion yield, ten randomly selected plots were harvested per treatment (biofertilized and non-biofertilized); each plot consisted of five rows of onion (10 m long, 30 cm row spacing).

### 2.4. Statistical Analysis

One-way ANOVA was used to analyze differences among the growth, quality, and productive variables of the biofertilized and control plots for every year. Significant differences (*p* < 0.05) among these variables were determined using Tukey’s test [29]. Statistical differences (*p* < 0.05) in the fungal incidence percentages between the biofertilized and non-biofertilized plots were analyzed using Fisher’s exact test [30].

### 2.5. Verification of the Presence of Beneficial Microorganisms Within Internal Onion Tissues

At the end of the growing cycle of the onion, 35 plants from the plot treated with the systemic biologicals were randomly collected to re-isolate the bacteria and fungi previously applied to the onion crop; the same quantity of plants were harvested from the control, non-inoculated plot. Samples from onions (leaves and bulbs) were washed under running water and cut into small fragments (2 × 2 cm). Subsequently, in a laminar flow hood, these pieces were disinfected with 70% ethanol for 1 min and 3% sodium hypochlorite for 2 min, washed thrice with sterile distilled water and then placed on sterile paper napkins to eliminate excess moisture. Finally, the onion fragments were cut into smaller fragments (1 × 1 cm) and placed in Petri dishes containing selective media for the isolation of *Bacillus subtilis* (BS medium) [31], *Azospirillum brasilense* (Congo Red and Elmarc media) [32], *Pseudomonas fluorescens* (Gould S1 medium) [33], and *Trichoderma* spp. [34]. The Petri dishes containing the onion pieces were incubated for 24 h at 28–30 °C. Morphologically distinct microbial colonies were selected and purified. *Trichoderma* strains were analyzed under a microscope to measure the size and morphology of conidia, conidiophores, mycelia, and other important fungal structures, using taxonomic identification keys specifically developed for our scientific group to identify and distinguish our *Trichoderma* strains. For bacteria, molecular analyses were carried out for identifying the specific bacterial strains inoculated on the onion plants.

### 2.6. DNA Extraction, PCR, and Restriction Analysis for Bacterial Strains

Briefly, the re-isolated microorganisms were grown in 10 mL of their respective growth media for 24 h. Then, the bacterial samples were centrifuged at 5000 rpm for 10 min. The supernatant was discarded, and the remaining pellets were used for genomic DNA extraction by the sarcosine method [35]. To verify DNA quality, electrophoresis was performed with 1% agarose gels. Using the extracted DNA as a template, fragments of the internal transcribed spacer (ITS) regions were amplified using the primers G1 (SEC ID NO: 7) and L1 (SEC ID NO:8) [15], in mixtures containing (in 50 μL final volumes) genomic DNA (150 ng), Buffer 1X, MgCl_2_ (2 mM), nucleotides (50 μM each), primers (0.2 μM each), and Taq polymerase (1 U per reaction). The mixtures were then placed in a Thermo Scientific thermocycler, considering the following conditions for amplifications: initial phase, 5 min at 94 °C for one cycle; denaturation at 94 °C for 1 min; annealing at 55 °C for 2 min; and extension at 72 °C for 2 min for 35 cycles.

A 7 min final extension at 72 °C for 1 cycle was performed at the end of the cycling steps. After amplification, samples were maintained at 4 °C. The amplicons obtained were purified using a QIAEX II kit (Qiagen, Hilden, Germany) and subsequently restricted using the enzyme *Dde* I (Table 2). Previously, we had generated DNA fingerprints of three bacteria based on the restriction patterns of the ITS fragments amplified using the restriction enzymes *Hae* III, *Dde* I, and *Hha* I.

## 3. Results

The prevailing climatic conditions during both years of the study were favorable for onion growth. Annual precipitations of 480.2 mm in 2022 (51.5 mm during the first productive cycle of the onion; Figure 1) and 453.3 in 2023 (113.5 mm during the second cycle of onion) are considered suitable for onion growth. However, to always maintain humidity at the levels required for onion maximum production, soil water potentials, as measured with soil irrometer tensiometers, were always maintained between −0.30 and −0.4 bars at a soil depth of 20 cm by means of the irrigation system. Temperature was also favorable for onion growth with December of 2022 showing an average temperature of 15.5 °C and May presenting an average temperature of 23.4 °C. During 2023, the coolest month was December (14.4 °C), while the warmest month was June (24.9 °C; Figure 1).

Performance of the systemic biologicals applied in onion in this study led to consistent and robust results from the initial samplings of plants carried out at 60 DAT to the final samplings at 120 DAT. Taken together, all of the variables analyzed in this study showed evident benefits of the systemic biologicals employed in this study on the onion performance (Table 3 and Table 4). Thus, with the use of the systemic products, plant height and weight were significantly increased by 35% and 58%, respectively, with relation to the conventionally managed plants in 2022 (Table 3; *p* < 0.5), and by 19% and 59% during 2023 (Table 4; *p* < 0.5). During the initial sampling, inoculated plants developed up to almost 3.6 more leaves than control plants in 2022, and 4.8 more leaves during 2023 (*p* < 0.5). At the end of the productive cycle of 2022 (120 DAT; Table 3) the inoculated plants were 17 cm taller, 313.2 g heavier, and developed 5.6 leaves more than their control counterparts (*p* < 0.5; Table 3).

Similarly, in 2023, onion plants inoculated with the systemic biologicals were significantly 26% taller, 66% heavier, and produced 48.3% more leaves than the control plants (*p* < 0.5; Table 4). Onion bulbs collected from the experimental plot at harvest time were significantly heavier (45.1% in 2022 and 56.2% in 2023) than their counterparts collected at the control, non-inoculated plot (*p* < 0.5; Table 3 and Table 4).

As with the other growth variables analyzed in this research, differences in chlorophyll accumulation between inoculated and non-inoculated onion plants were evident at all sampling dates and during both years of the study (Table 3 and Table 4). At the end of the 2022 experiment, plants collected at the experimental plot developed 23.5% more chlorophyll (41.9 µg∙g FW^−1^) than plants collected at the control plot (*p* < 0.5; Table 3). In the same way, plants inoculated with the systemic biological products during 2023 developed 41.7% more chlorophyll than control plants (231.4 vs. 163.3 µg∙g FW^−1^, *p* < 0.5).

As expected, all of the differences observed in the variables considered in this research and in both years ultimately resulted in a higher onion yield in the plot inoculated with the systemic products than in the conventionally managed plot. During 2022, the experimental plot produced 14.4 t∙ha^−1^ more than the control plot (*p* < 0.5), whereas in 2023, this difference was greater, i.e., 19.4 t∙ha^−1^ in favor of the inoculated plot (*p* < 0.5); the percentage differences in the yields of the two years evaluated showed an average increase of 44.4%.

On the other hand, onion quality expressed in terms of total soluble solids and pyruvic acid contents was significantly greater (*p* < 0.5) in the onion plants inoculated with the biological products over the two-year study period in comparison with the plants not receiving the plant growth-promoting microorganisms (Table 5). The two-year average values for these variables were 10.2 vs. 14.4°Brix and 2.3 vs. 4.0 μmol∙g^−1^ for the control and the biologically treated plants, respectively.

With respect to the fungal disease occurrence in the onion crop in both years, it could be determined that fungal incidence at harvest time in the two years of evaluation was statistically lower in the plot where the systemic biologicals were applied (*p* < 0.5; Table 5); the fungal incidence was 1.7 and 2.1 times greater in the control than in the inoculated plants during 2022 and 2023, respectively. The fungal pathogens attacking the onion crop were identified as *Sclerotium cepivorum* and *Fusarium oxysporum.*

Finally, the analysis of the onion plants selected from each, the biologically inoculated plot and the control, non-inoculated plot, by morphological and molecular techniques, revealed the presence of at least two of the four strains of *Trichoderma* included in the product Trichonator^®^ and the three bacterial species contained in BactoCROP^®^ within the internal tissues of the onion plants (Table 6). These microorganisms were not detected in the conventionally managed plants.

## 4. Discussion

The data presented in this study demonstrate a robust and consistent performance of the systemic biological products employed in the onion. The functionality of the biofertilizer and the biofungicide/biofertilizer on the onion performance was evident since the first applications. In our experience, one of the first signals of proper functioning of the systemic biologicals in the onion, as in some conventional biological [36] and organic products [12] employed in the onion, is a significant increase in chlorophyll content [11].

After this chlorophyll increase in onion plants, improvement in the growth variables analyzed in this study revealed the benefits of applying the systemic products on onion plants at all sampling dates. Plant height and weight, number of leaves, and bulb weight were significantly greater (*p* < 0.5) in onion plants inoculated with the systemic biologicals (Table 3 and Table 4) than in control plants.

These results are in accordance with those of other researchers evaluating the application of biological products on onions. Gemin and collaborators, in 2019 [37], successfully increased the caliber of onion bulbs and, consequently, the yield of this crop by 40% in the onion cultivar ‘Perfecta F1’ through inoculation of the microalgae *Scenedemus subspicatus* (30 control vs. 42 t∙ha^−1^ inoculated). Conversely, in other onion cultivar evaluated in this study (‘BR-29’), no differences were detected between the control and the experimental plot.

Cordeiro and collaborators, in 2022 [36], applying the microalgae *Asterarcys quadricellulare* at different concentrations on two onion cultivars, namely ‘Alvará’ and ‘Perfecta’, observed statistically significant differences in diverse productive variables of onion in comparison with the control treatment. As a consequence of producing larger and more uniform bulbs with the inoculation of the microalgae, increases in onion yield were detected in both onion cultivars. In ‘Alvará’, a maximal increase in the onion yield of 28.3% was obtained (30 vs. 41 t∙ha^−1^), whereas in ‘Perfecta’, the greatest increase was 40.0%, as compared to the control treatment (28 vs. 39 t∙ha^−1^).

In 2014, Čolo and collaborators [38] evaluated the application of *Azotobacter chroococcum*, *Bacillus subtilis,* and *Pseudomonas fluorescens*, and the combination of these microorganisms, on the growth and yield of onion. Inoculation of *A. chroococcum* and *B. subtilis* resulted in longer seedlings, whereas plant height was greater in individuals treated with any of the bacteria evaluated (sixty days after inoculation). At the end of the productive cycle of the onion, yield was higher in plants inoculated with *A. chroococcum* (34.73 t∙ha^−1^, 97% increase) or with *B. subtilis* (28.8 t∙ha^−1^, 63.5% increase) than in non-inoculated plants (17.60 t∙ha^−1^), i.e., the average of the biological treatment increased by 80% the onion yield with relation to the control treatment.

Singh and colleagues, in 2017 [39], compared different combinations of chemical fertilization doses with biological inoculants including *Azospirillum*, VAM (vesicular-arbuscular mycorrhizae), and a PSB (phosphate solubilizing bacterium). Their findings revealed that the complete fertilization dose of 120-60-80 combined with *Azospirillum* and VAM resulted in the maximum plant height (51.96 cm), number of leaves per plant (11.96), leaf length (45.71 cm), fresh weight of leaves (32.58 g), and bulb yield (46.8 t∙ha^−1^) in comparison with the control treatment consisting in the complete dose of fertilization. Similarly, onion yield components such as bulb length (5.13 cm), bulb diameter (5.85 cm), bulb weight (81.44 g), and bulb volume (92.8 cc) were greater in this treatment. Bulb yield was increased by 23% in the biologically treated onions with relation to the control counterpart group (38.3 vs. 46.8 t∙ha^−1^).

Arunachalam and collaborators, in 2024 [40], evaluated the effect of applying PSB and mycorrhizae on the growth and nutrient uptake of onions cultivated during two different seasons (2018–2019 and 2019–2020). The evaluated plant growth-promoting microorganisms included *Serendipita indica* (a fungal species of the *Basidiomycota* Division and *Sebacinales* order), a PSB consortia consisting of *Bacillus megaterium*, *Paenibacillus polymyxa* and other *Bacillus* sp., and a VAM consortia consisting of *Glomus fasciculum*, *G. intraradices*, *Acaulospora* sp., and *Gigaspora* sp. These microorganisms were combined with 50% (50% RDF) or 100% (100% RDF) of the recommended dose of chemical fertilization. Their results indicated that 100% RDF combined with *S. indica* or the PSB consortia led to improved plant growth, and higher nutrient concentrations in both the leaves and bulbs of onion compared to the 100% RDF treatment alone. Moreover, the application of 100% RDF with *S. indica* increased the total dry matter yield by 11.5% and 7.6% in the 2018–2019 and 2019–2020 seasons, respectively, in comparison with the complete recommended dose of fertilization (100% RDF). This treatment also resulted in the greatest nutrient assimilation, with N uptake increasing during the two seasons (2018–2019 and 2019–2020) by 6.9% and 29.9%, P by 13.7% and 21.7%, K by 20.0% and 23.7%, and S by 18.1% and 23.4%, respectively, with comparison to the treatment consisting in the complete chemical fertilization (100% RDF). The combination of 100% RDF with *S. indica* inoculation led to a significant increase in bulb yield in comparison with the 100% RDF treatment alone only in the first season (40 vs. 35 t∙ha^−1^, 14.2% increase). The yield registered in the treatment including *S. indica* plus 100% RDF was also significantly greater during both seasons in comparison with the onion yield observed in treatments receiving only 50% RDF or 0% RDF (absolute control).

Manna and collaborators, in 2014 [41], investigated the interactive effects of applying chemical and biological fertilizers on the growth, yield, and quality of the onion. They reported that the maximum plant height (67.66 cm), bulb diameter (4.82 cm), bulb weight (56.92 g), total yield (30.20 t∙ha^−1^), and marketable yield (28.43 t∙ha^−1^) were recorded with the application of the complete recommended dose of chemical fertilizers plus the addition of a phosphate-solubilizing bacterium and *Azospirillum*. Total onion yield in this biological treatment was 42.5% greater than that registered in the control treatment (21.03 t∙ha^−1^), consisting in the use of the complete recommended dose of chemical fertilizers.

On the basis of the referred studies, we can conclude that increases in the yield of onions using conventional biological products and the complete recommended dose of chemical fertilization are highly variable: 97% with the use of *Azotobacter chroococcum* or 63.5% with *Bacillus subtilis* [38]; 23% using a phosphate-solubilizing bacterium and 64.2% with a mix of *Azospirillum* and VAM [39]; 14.2% with *Serendipita indica* [40]; and 42.5% using a phosphate-solubilizing bacterium plus *Azospirillum* [41].

On the other hand, it is very important to state that there is no guarantee that increases in onion yield over the traditional management of crops should be always expected when conventional biological products are employed. Kadam and collaborators, in 2023 [42], were unable to reach the yield obtained with the complete (22.2 t∙ha^−1^) or half (18.4 t∙ha^−1^) of the recommended chemical fertilization dose for onions in India (100-50-100) by inoculating different combinations of *Azospirillum*, *Azotobacter*, a VAM, and a phosphate-solubilizing bacterium. The more efficient biological combination was obtained when *Azospirillum* and *Azotobacter* were jointly inoculated on the onion (18.1 t∙ha^−1^).

Using systemic biological products, in the present study, we were able to consistently and significantly increase (*p* < 0.5) the onion yield in 14.4 (42.5% increase), and 19.4 t∙ha^−1^ (46.3% increase), the first and the second year of the study, respectively. The onion yields reached in our study by using systemic biologicals in the cultivar ‘Carta Blanca’ are well above the average onion yield in Mexico, which is around 31 t∙ha^−1^ [8].

We confirmed the positive interaction of chemical fertilizers with (systemic) biologicals in the onion cultivation. These data agree with previous studies carried out by us on sugarcane, where increases in yield were very consistent and reliable over six years [15].

Although Petrovic and collaborators, in 2019 [19], reported different advantages in using products containing organic matter (such as manure) or vermicompost over the application of biological products on the yield and different growth parameters of the onion, advantageous synergies of conventional and systemic biologicals in combination with organic nutritional sources are expected in the cultivation of agricultural crops, particularly the onion.

Verma and collaborators, in 2021 [43], evaluated the application of different beneficial microorganisms combined with vermicompost on the growth and yield of onion. Bulb yield was greater in onion plants cultivated with the complete chemical fertilization and inoculated with a combination of beneficial microorganisms (*Azospirillum*, VAM, and a phosphate-solubilizing bacterium) and vermicompost than in control plot managed only with the complete chemical fertilization. A maximal increase of 18% in the onion yield by effect of the biological/organic treatment was reported (30.2 t∙ha^−1^ in the experimental plot vs. 25.6 t∙ha^−1^ in the control plot).

Sarhan and Bashandy, in 2021 [44], analyzed the effects of applying manure, copper, and *Azobacter chroococcum* on the yield and quality of onion. The highest yields were obtained in plots where 24 t∙ha^−1^ of manure was applied, regardless of the inoculation with *A. chroococcum* or the foliar spray with copper sulphate. Onion yield in plots managed with 24 t∙ha^−1^ of manure was between 50 and 52 t∙ha^−1^ with no statistical differences between plots with or without both copper and *A. chroococcum*. Moreover, plots managed with 12 t∙ha^−1^ of manure plus copper and *A. chroococcum* showed no statistical differences with plots receiving 24 t∙ha^−1^ manure.

Accordingly, as in the case of chemical nutritional sources, increment expectations in onion yield by the use of biological products in the presence of organic sources are also quite variable: 40% using *Scenedesmus subspicatus*; 28.3 to 40% with *Asterarcys quadricellulare*; and 18% with a mix of *Azospirillum*, *Azotobacter*, *Bacillus*, *Herbaspirillum,* and *Chlorella vulgaris*.

Moreover, reductions in the organic fertilization doses are possible with the use of microorganisms. For example, manure doses can be reduced by half in the presence of *Azobacter chroococcum* without losses in potential onion yields [44].

The benefits of using plant growth-promoting microorganisms extend beyond increasing the yield of onion. Different studies have shown improvements in different quality variables of this crop by using beneficial microorganisms. A study on the enhancement of onion quality (antioxidant activity) by effect of applying a biofertilizer containing *Azospirillum*, *Bacillus*, *Chlorella vulgaris,* and *Herbaspirillum* was published by Petrovic and collaborators in 2020 [12].

Similarly, improvements in reducing sugars (%) and non-reducing sugars (%) have also been described in onion plants inoculated with VAM compared with control, non-inoculated plants fertilized with the complete dose of NPK [45].

Manna and collaborators, in 2014 [41], found that the quality of the onion bulbs, in terms of total soluble solids and pyruvic acid contents, was improved with a biological treatment including *Azospirillum* and a phosphate-solubilizing bacterium, whereas Sarhan and Bashandy, in 2021 [44], concluded that fertilizing the onion plants with 12 t∙ha^−1^ manure plus copper sulphate and inoculating them with a biofertilizer makes it possible to reach the same improvements in onion quality (such as bulb diameter, total soluble solids, and onion storability) with the application of 24 t∙ha^−1^ manure, with obvious repercussions in the net economic returns and benefit/cost ratio.

In addition to obtaining significative increases in onion yields by using systemic biologicals in this study, we were able to enhance the quality parameters of onion, particularly the contents of pyruvic acid and total soluble solids. Pyruvic acid concentration directly correlates with the onion’s pungency; this is due to the enzymatic production of pyruvic acid as a byproduct when onion cells are ruptured, releasing the compounds responsible for the characteristic onion flavor and tear-inducing effect. On the other hand, the content of total soluble solids in onions is important because it directly impacts the onion’s storage life, flavor intensity, and overall quality, with higher total soluble solids generally indicating better storage potential and a milder, sweeter taste due to increased sugar content; essentially, onions with higher total soluble solids tend to last longer in storage without significant quality degradation [46,47].

In the present research, the biological treatments were imposed over the local conventional treatment of onion; only the chemical fungicide methyl thiophanate was omitted from the experimental plot inoculated with the beneficial microorganisms. The results obtained in this study reflect the particular activities of the microorganisms on the nutrition and health of onion. Firstly, the results registered since the initial samplings on the plant growth variables measured in onions clearly show the plant growth promotion activities of the microorganisms included in BactoCROP^®^, i.e., *Pseudomonas*, *Bacillus,* and *Azospirillum*. Previously, we had demonstrated the effects of this systemic biofertilizer on the growth and production of other vegetables and cereals [11]. Plant growth promotion activities of the microorganisms included in the formulation of the biofertilizer employed in this study include nitrogen fixation, phosphate solubilization, hormone production, ACC deaminase activity, and siderophore production, among others [11]. However, we cannot rule out a possible role of the biofungicide on the nutrition of the onion plants, because some of the strains included in Trichonator^®^ have phosphate-solubilizing capabilities.

These results obtained in onions utilizing systemic biological products jointly with complete chemical fertilization has also been described by us in other crops substituting the nutritional synthetic sources by organic sources, such as chicken or cow manure [11].

Another factor contributing to the improved growth of onion plants was the relatively low levels of pathogenic fungi detected in the inoculated plants in comparison with those observed in the control plants, which indicated the control efficiency exerted by the *Trichoderma* strains contained in Trichonator^®^. According to previous studies, the application of plant growth-promoting microorganisms along with mineral fertilizers enhances plant growth and crop yield by mechanisms including phytohormone secretion, nutrient supplementation, and pathogen suppression [48,49]. In other words, a better nourished plant can better resist the attack of pathogens [50].

We visualize two approaches for the application of biological products in agriculture. The first one is focused on rainfed crops, in which the minimization of costs is central, given the possibility that the occurrence of suboptimal precipitations or temperatures, or the presence of plagues or diseases, significantly compromises the potential yields of crops. The second one is directed to vegetables, fruit trees, sugarcane, and other highly profitable crops, in which the search for the highest yields is the main goal.

## 5. Conclusions

Results presented in this research confirm our previous findings related to a better and consistent functioning of the systemic biological products in agricultural crops in comparison with conventional biologicals. Although a direct comparison between systemic and conventional biological products was not contemplated, onion yields obtained in this research using systemic biologicals were exceptional for the common yields registered in Mexico using the cultivar ‘Carta Blanca’. Also, the differences found by us between the control and the experimental plots were extraordinary if we compare them with the published literature in onion inoculated with conventional biological products. Because systemic biological products have proven to be consistent and efficiently increase the yield and quality of crops, major efforts should be devoted to improving our knowledge on the functioning of these biological products. This is particularly relevant if we consider that systemic biologicals are the only products that can be successfully applied in agricultural systems such as waterlogged rice, aquaculture, or mangrove ecosystem restoration as well as in the employment of agricultural drones for inoculating beneficial microorganisms on the field.

## Figures and Tables

**Figure 1 microorganisms-13-00797-f001:**
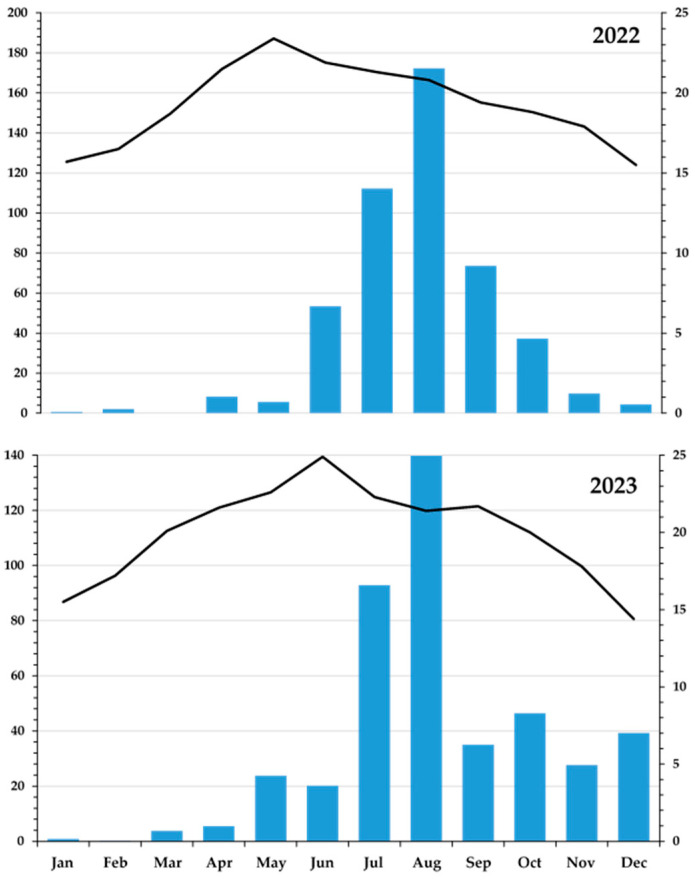
Climographs of San Juan de la Vega, Guanajuato, Mexico, showing the precipitation (bars) and the average temperature (line) during the years of 2022 and 2023.

**Table 1 microorganisms-13-00797-t001:** Composition of the vehicle used for formulating the two biological products applied in this study, BactoCROP^®^ and Trichonator^®^.

Component	%
Protein	9.3
Polysaccharides	8.2
Carbohydrates	9.3
Phosphorous	0.7
Potassium	1.2
Iron	1.9
Calcium	0.5
Magnesium	0.5

**Table 2 microorganisms-13-00797-t002:** Expected band sizes (bp) by the amplification of the ITS region and its restriction patterns (bp) generated with the enzyme *Dde* I in the three bacteria.

	Bacterial Species		
	*P. fluorescens*	*A. brasilense*	*B. subtilis*
Amplified fragments from ITS region	692 654 614	668	455 265
Restriction fragments generated with *Dde* I	195 138 105	190 122 88	208

**Table 3 microorganisms-13-00797-t003:** Productive variables analyzed in biofertilized and conventionally managed onion plots located at San Juan de la Vega, Guanajuato, Mexico, during 2022.

Variable	Days After Transplantation					
	60		90		120	
	Control	Inoculated	Control	Inoculated	Control	Inoculated
Plant height (cm)	37.1 a	50.1 b	38.9 a	78.8 b	83.1 a	100.1 b
Plant fresh weight (g)	239.8 a	379.6 b	399.9 a	512.6 b	467.9 a	781.1 b
No. leaves	6.2 a	9.8 b	7.1 a	10.6 b	10.2 a	15.8 b
Bulb fresh weight (g)	68.9 a	113.0 b	188.7 a	289.3 b	357.9 a	519.4 b
Chlorophyll (µg∙g FW^−1^)	111.2 a	137.7 b	121.6 a	137.2 b	178.3 a	220.2 b

Average values of 25 replicates. Rows with the same letter are not significantly different between conventionally managed (control) and inoculated plots for each transplant date, as determined by Tukey’s mean separation (*p* > 0.05).

**Table 4 microorganisms-13-00797-t004:** Productive variables analyzed in biofertilized and conventionally managed onion plots located at San Juan de la Vega, Guanajuato, Mexico, during 2023.

Variable	Days After Transplantation					
	60		90		120	
	Control	Inoculated	Control	Inoculated	Control	Inoculated
Plant height (cm)	47.3 a	56.1 b	60.3 a	86.2 b	78.9 a	99.8 b
Plant weight (g)	279.8 a	444.3 b	411.1 a	554.8 b	486.6 a	811.3 b
No. leaves	7.2 a	12.0 b	8.9 a	14.8 b	11.6 a	17.2 b
Bulb weight (g)	82.4 a	115.5 b	200.8 a	330.8 b	366.8 a	572.8 b
Chlorophyll (µg∙g FW^−1^)	109.1 a	138.0 b	122.3 a	157.9 b	163.3 a	231.4 b

Average values of 25 replicates. Rows with the same letter are not significantly different between conventionally managed (control) and inoculated plots for each transplant date, as determined by Tukey’s mean separation (*p* > 0.05).

**Table 5 microorganisms-13-00797-t005:** Yield, quality, and fungal incidence in onion plants inoculated and non-inoculated with systemic biologicals.

Variable	2022		2023	
	Control	Inoculated	Control	Inoculated
Yield (t∙ha^−1^)	33.8 a	48.2 b	41.9 a	61.3 b
Total soluble solids (°Brix)	10.21 a	14.15 b	10.30 a	14.62 b
Pyruvic acid content (μmol/g)	2.24 a	3.78 b	2.45 a	4.22 b
Fungal incidence (%)	16.1 a	9.5 b	18.7 a	9.1 b

Average values of 10 (yield) and 25 replicates (onion quality variables and fungal incidence). Rows with the same letter are not significantly different between conventionally managed (control) and inoculated plots for each transplant date, as determined by Tukey’s mean separation (*p* > 0.05) for the yield and quality variables, and by Fisher’s exact test for the fungal incidence percentages.

**Table 6 microorganisms-13-00797-t006:** Frequency of the inoculated microorganisms within the onion plants.

Microorganism	Frequency (%)
*Azospirillum brasilense*	100
*Bacillus subtilis*	100
*Pseudomonas fluorescens*	100
*Trichoderma hamatum*	39
*T. harzianum*	55
*T. koningii*	53
*T. viridae*	52

## Data Availability

Data are contained within the article.

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
