# Peer review of "Growth, Health, Quality, and Production of Onions (Allium cepa L.) Inoculated with Systemic Biological Products"

_microorganisms, 2025, doi:10.3390/microorganisms13040797_

Round 1

Reviewer 1 Report

Comments and Suggestions for Authors

Gutiérrez-Benicio et al., provide valuable insights on growth, quality and production of onion (Allium cepa L.) inoculated with systemic biological products. The manuscript is well written; it is mainly focused on onion crop growth and protection.

There are some points to be mentioned:

  1. Title is crop oriented and not oriented towards microorganisms as it may be appropriate; plant health of onions was not challenged in this work as the title mentions.
  2. Please provide some more information for the term “systemic” in “systemic biological products” word sequence; it may be confused with systemic agrochemicals for plant use.
  3. Extensive part of introduction is related to onion crop farm practices, onion as local and global commodity concluding to list of onion pathogens (lines 32-76, 73-115); biofertilizers are mentioned within introduction but the focused-approach on microorganisms in this study is extremely limited. Authors must introduce the reader to the piece of work they present in the article; major changes of introduction’s content need to take place.
  4. Data sheet / label of BactoCROP from the manufacturing company (https://www.bioqualitum.com/data/productos/BactoCROP/ficha.pdf ) mentions that the product encloses only two microbial species (Bacillus and Azospirillum), see relative text  [Biofertilizante compuesto por un consorcio de bacterias benéficas de los géneros Bacillus y Azospirillum que contribuye al aumento de la productividad de los cultivos….]. The data sheet/label does not report that Pseudomonas fluorescens is included in the product as mentioned in the manuscript. It would be nice to provide some more information or a relative reference.
  5. Table 1 refers to composition of the vehicle used for formulating the two biological products applied in this study, BactoCROP® and Trichonator®. Please define if this information is label-based or it is additional bench work done by the authors.
  6. Please provide information why was chosen to: a) re-isolate and perform DNA extraction, PCR, and restriction analysis for the bacterial strains (line 218) and b) to not isolate samples and perform similar genetic analysis for Trichoderma applied species. Presence of microorganisms in the field is not coupled with quantification data; it is hard to make a fully conclusive statement for a) sustainable and/or resilient microbial-based practice b) the fate of microorganisms.
  7. Table 2 provides expected band sizes (bp) for the three bacteria; however, no data from microbial presence checks are demonstrated in result’s section. Please provide relative information.
  8. The way that the experimental plan was set, it permits to create a conclusive statement for the two jointly applied formulated products. Although, no comparative or conclusive statements per microbial type, group or species can be documented.

Reviewer 2 Report

Comments and Suggestions for Authors

In my opinion, this study on the effects of using different microorganisms to improve onion crop yield is interesting. Nevertheless, the application of different types of products together or different species makes it difficult to know the real individual effect of each one. Doses of the different microorganisms and species should be included and the hypothesised individual effect should be discussed in comparison with the bibliography. The research seems a little short in terms of the different variables studied and the analysis shown, that's why I suggest to include some additional statistical analysis. Several aspects of the manuscript should be improved, such as the introduction, which should focus on the background of this research. Materials and methods should also be improved with some additional details. The results as reported seem to be a little short although the evidence of significance in the parameters studied is there. Conclusions should be added or preferably separated from the discussion. Additional comments are provided in the attached manuscript. Acceptance at the discretion of the editor.

Comments on the Quality of English Language

English should be reviewed by a native speaker.

Reviewer 3 Report

Comments and Suggestions for Authors

The manuscript, titled "Growth, health, quality and production of onion (Allium cepa L.) inoculated with systemic biological products", presents the results of treating an important vegetable species with systemic biological pesticides compared to conventional cultivation methods. In this research they evaluated the functioning of two commercial systemic biological products (BactoCROP and Trichonator) on the growth, quality and yield of onion. BactoCROP contains Pseudomonas fluorescens, Azospirillum brasilense, and Bacillus subtilis as active ingredients, whereas Trichonator contains four different Trichoderma species (Trichoderma viridae, T. harzianum, T. koningii and T. hamatum).

The introduction is adequate and sufficiently detailed. The materials and methods chapter is sufficiently detailed and well describes the materials, methods used, and the Mexican study site. I recommend a uniform, italicized spelling of scientific names throughout the whole manuscript.

I also recommend reporting which pathogens were detected during the determination of the Fungal incidence %!

The presentation of the results illustrates the achieved results well-explained and well-edited figures and tables used for clear presentation.

The conclusion chapter makes comparisons with several related studies. The number of referenced scientific works is adequate (46) and provides a sufficient basis for comparison for evaluating own results.

After implementing the suggested corrections and additions, I recommend publishing the manuscript as a scientific article.

Round 2

Reviewer 1 Report

Comments and Suggestions for Authors

Proposed changes/suggestions applied to the manuscript. 

Reviewer 2 Report

Comments and Suggestions for Authors

Dear Authors, Thank you for answering my questions. I consider the research is now publishable.

Kind regards.